# eYGFPuv-Assisted Transgenic Selection in *Populus deltoides* WV94 and Multiplex Genome Editing in Protoplasts of *P. trichocarpa* × *P. deltoides* Clone ‘52-225’

**DOI:** 10.3390/plants12081657

**Published:** 2023-04-14

**Authors:** Guoliang Yuan, Yang Liu, Tao Yao, Wellington Muchero, Jin-Gui Chen, Gerald A. Tuskan, Xiaohan Yang

**Affiliations:** 1Oak Ridge National Laboratory, Biosciences Division, Oak Ridge, TN 37831, USA; 2The Center for Bioenergy Innovation, Oak Ridge National Laboratory, Oak Ridge, TN 37831, USA; 3Chemical and Biological Process Development Group, Pacific Northwest National Laboratory, 902 Battelle Boulevard, Richland, WA 99352, USA

**Keywords:** eYGFPuv, transformant screening, CRISPR-Cas9, gene editing, transient transformation, protoplasts

## Abstract

Although CRISPR/Cas-based genome editing has been widely used for plant genetic engineering, its application in the genetic improvement of trees has been limited, partly because of challenges in *Agrobacterium*-mediated transformation. As an important model for poplar genomics and biotechnology research, eastern cottonwood (*Populus deltoides*) clone WV94 can be transformed by *A*. *tumefaciens*, but several challenges remain unresolved, including the relatively low transformation efficiency and the relatively high rate of false positives from antibiotic-based selection of transgenic events. Moreover, the efficacy of CRISPR-Cas system has not been explored in *P*. *deltoides* yet. Here, we first optimized the protocol for *Agrobacterium*-mediated stable transformation in *P*. *deltoides* WV94 and applied a UV-visible reporter called eYGFPuv in transformation. Our results showed that the transgenic events in the early stage of transformation could be easily recognized and counted in a non-invasive manner to narrow down the number of regenerated shoots for further molecular characterization (at the DNA or mRNA level) using PCR. We found that approximately 8.7% of explants regenerated transgenic shoots with green fluorescence within two months. Next, we examined the efficacy of multiplex CRISPR-based genome editing in the protoplasts derived from *P*. *deltoides* WV94 and hybrid poplar clone ‘52-225’ (*P. trichocarpa* × *P. deltoides* clone ‘52-225’). The two constructs expressing the Trex2-Cas9 system resulted in mutation efficiency ranging from 31% to 57% in hybrid poplar clone 52-225, but no editing events were observed in *P*. *deltoides* WV94 transient assay. The eYGFPuv-assisted plant transformation and genome editing approach demonstrated in this study has great potential for accelerating the genome editing-based breeding process in poplar and other non-model plants species and point to the need for additional CRISPR work in *P. deltoides*.

## 1. Introduction

*Populus* is commonly recognized as the ‘model’ woody plant for genomics research due to fast growth rate, vegetative propagation, relatively small genome size, the large number of molecular tools, and the ease of genetic transformation [1,2]. More recently, the use of *Populus* spp. as a biomass feedstock has also been widely highlighted for generating sustainable and renewable transportation fuels with chemical properties comparable to those of present gasoline, diesel, and jet fuel [1,3,4]. Developing a variety of genetic and genomic resources are a prerequisite for accelerating the domestication of *Populus* to meet the growing demand on the improvement of the wood quality and yield from the industry. Since the genome sequence of *P. trichocarpa* was published in 2006 [2], the genomes of multiple other *Populus* species have been sequenced and annotated, including *P. euphratica* [5], *P. pruinosa* [6], *P. alba* [7,8], *P. simonii* [9], and *P. deltoides* [10]. Notably, these species cannot be easily transformed by *Agrobacterium tumefaciens* [11]. Generally, only hybrid poplar species, which have much higher transformation rates, have been widely used for genetic transformation, e.g., *P*. *tremula* × *P*. *alba* clone ‘717-1B4’ [12], *P. alba* × *P. tremula* var. glandulosa clone ‘84K’ [13], *P. alba* × *P. grandidentata* [14], and *P. alba* × *P. tremula* [15].

*Populus deltoides* (eastern cottonwood) is naturally and widely distributed in the United States and southern Canada [16], and its genetic resources have been used as the primary gene donors for the development of poplar cultivars [17]. For example, interspecific hybrids of *P. trichocarpa* × *P. deltoides* have been used for identifying genome regions of ecological and adaptive traits [18]. The clone 52-225 (*P. trichocarpa* × *P. deltoides* ‘52-225’) has been identified as a strong candidate for bioenergy and wood production [19,20]. A single *P. deltoides* clone ‘WV94’, from Issaquena Co., Mississippi, first identified by the U.S. Forest Service because of its rapid growth under field conditions [21], was selected by ArborGen, Summerville, SC, USA for use in its transformation program. Up to now, *Agrobacterium*-mediated transformation has been successfully using *P. deltoides* clone WV94 [22,23,24]. The genome sequence of *P. deltoides* WV94 is now publicly available in the Phytozome database http://phytozome.jgi.doe.gov (accessed on 9 December 2022). However, increased transformation efficiency or an easy and rapid screening method remains to be developed for *P. deltoides* in general and WV94 specifically.

The CRISPR/Cas9 system has revolutionized genome engineering in plants because of its simplicity and efficiency [25,26,27]. However, in *Populus*, CRISPR/Cas9-mediated genome engineering has only been achieved in limited poplar genotypes, such as, *P. davidiana* × *P. bolleana*, *P. tremula* × *P*. *alba* clone INRA 717-1B4, and *Populus tomentosa* [28,29,30,31]. nCas9-based base editing has been used to achieve 100% editing frequency with PmCDA1-BE3 in *P. tremula* × *P*. *alba* 717-1B4 [28] and the CRISPR combo system has been used to activate the poplar morphogenic gene WUSCHEL (*PtWUS*) to accelerate regeneration in *P. davidiana* × *P. bolleana* [29].

Beyond basic CRISPR/Cas9 genome editing, co-expression of *Trex2* exonuclease with Cas9 has been shown to enhance the efficiency of CRISPR/Cas9 mutagenesis in plants [32]. In addition, geminivirus replicons (GVRs) have been used for increasing donor template availability by replicating it to high copy number and hereby increasing the CRISPR/Cas-mediated gene targeting efficiency [33]. However, due to the lack of a stable transformation system, application of the CRISPR/Cas9 system in other poplar genotypes has been limited and there are no reports on GVRs and Trex2-Cas9 system in poplar. Here, we tested the *Bean yellow dwarf virus* (BeYDV) replicons and Trex2-Cas9 system for multiplexing genome editing in both WV94 and 52-225 protoplasts.

Previously, we reported that a UV-visible reporter called eYGFPuv, which is an enhanced yellow GFP-like protein derived from the marine copepod *Chiridius poppei*, can be used as an indicator of gene expression in stable transformation of various plant species including *P*. *tremula* × *P*. *alba* 717-1B4 [34,35]. One of the key features of this reporter gene is that it enables early and efficient selection of transformants in a rapid and noninvasive manner [34]. In this study, we developed a simple and reliable method for early screening of transgenic events in *Agrobacterium*-mediated transformation of WV94 through the application of the eYGFPuv reporter. Furthermore, we achieved multiplexing genome editing using BeYDV replicons and Trex2-Cas9 system in the protoplasts of 52-225.

## 2. Results

### 2.1. eYGFPuv-Assisted Plant Transformation in Populus Deltoides

The stable transformation of *P. deltoides* WV94 typically involves seven steps: (1) preculture excised explants (petioles and base of primary vein) on co-culture medium 94 (CCM94) in the dark for four days, (2) incubate the precultured explants in *Agrobacterium* solution for one hour, (3) transfer the explants from *Agrobacterium* solution to CCM94 and keep it in the dark for three days, (4) wash the explants in wash solution containing 200 ng/μL Timentin and 300 ng/μL Cefotaxime for one hour, (5) transfer the explants from wash solution to shoot induction medium 94 (SIM94) and culture for two to three weeks, (6) transfer the explants from SIM94 to shoot elongation medium 94 (SEM94) and culture for two to three weeks, and (7) excise the shoots and transfer them to root medium 94 (RM94) and culture for two to three weeks or until roots are visible (Figure 1A). In comparison with the transformation of *P*. *tremula* × *P*. *alba* 717-1B4 [34], we found that transformation of WV94 is limited by three events. First, WV94 is more prone to *Agrobacterium* overgrowth during transformation. Second, the root induction of WV94 transgenic shoot is more difficult with a relatively low rate. Third, WV94’s relatively low transformation efficiency.

To prohibit the *Agrobacterium* overgrowth, the co-culture of explants and *Agrobacterium* on CCM94 in step (3) cannot exceed three days. Moreover, 200 ng/μL Timentin + 300 ng/μL Cefotaxime are necessary in all the selection media including SIM94, SEM94, and RM94. Furthermore, all the selection media containing antibiotics needs to be refreshed every 2–3 weeks. In addition, dipping the base of shoots into autoclaved Hormodin 2 (OHP, Inc) powder is required for a higher root induction rate. The addition of acetosyringone to the *Agrobacterium* cell culture in step (2) is also required to induce the expression of virulence genes in *Agrobacterium* required for plant genetic transformation [36].

Further efforts are also needed to rule out the false positive events through PCR genotyping in different stages of plant transformation, i.e., SIM94, SEM94, and RM94. As noted earlier, we recently demonstrated that the expression of the *eYGFPuv* reporter gene driven by 1 × 35S promoter can be used for early selection of transgenic events in the stable transformation of *P*. *tremula* × *P*. *alba* 717-1B4. It has also been reported that the 35S promoter with a duplicated enhancer (i.e., 2 × 35S) resulted in a higher GFP expression level in comparison with 1 × 35S promoter [37]. We thus evaluated a 2 × 35S promoter to drive *eYGFPuv* gene in this study (Figure 1B).

The eYGFPuv/GUS dual reporter vector was transformed into WV94 following the procedures described in Figure 1A. We then monitored the *eYGFPuv* expression from the shoot induction stage to the root induction stage. We observed bright green fluorescence on the transgenic explants under UV light as early as in the shoot induction stage (Figure 1C). Additionally, the signals of eYGFPuv were visualized continually through the shoot elongation stage to root induction stage (Figure 1C). Notably, explants without *eYGFPuv* expression exhibited red autofluorescence from chlorophyll under UV light, facilitating the identification of false positive events (Figure 1C). We also examined *eYGFPuv* expression of transgenic plants in the growth chamber and greenhouse. As expected, vivid green fluorescence was observed both six and nine weeks after plants were transferred to soil and no morphological phenotypes were observed in transgenic events (Figure 1D,E).

### 2.2. Verification of Transgenic Events

Because the dual reporter vector contained a GUS reporter, we used GUS staining to verify the transgenic events. Indeed, the leaf tissues with green fluorescence emitted from eYGFPuv in the shoot elongation stage stained blue (Figure 2A). Similarly, the leaf of nine-week-old transgenic event 1 in soil was also stained blue whereas no staining was observed in the wild type (Figure 2B). Finally, PCR genotyping was used to further confirm the positive transgenic events. Target bands were detected in all three randomly selected transgenic events (Figure 2C), indicating that all the transgenic events based on eYGFPuv-based selection were true positives.

### 2.3. Quantification of Transformation Efficiency

We performed quantitative analysis of transformation efficiency starting with 150 explants. We found that 143 out of the 150 explants (95.3%) survived on SIM94 containing 100 mg/L kanamycin in the shoot selection stage (Figure 3A and Table 1), with at least one shoot observed on 80 explants (53.3%) (Figure 3B). Based on the expression of *eYGFPuv*, we detected green fluorescence on 23 explants (15.3%) (Figure 3C and Table 1). In the shoot elongation stage, we observed 13 explants (8.7%) with at least one GFP shoot (Figure 3D and Table 1).

### 2.4. Development of Multiplexed Gene Editing Using Poplar Protoplasts

We used a protoplast-based assay for quickly assessing the Trex2-Cas9 system in poplar. To test if the Trex2-Cas9 could enable targeted mutagenesis in poplar, we designed two guide RNAs (gRNAs) targeting the conserved regions of the *phytoene desaturase* (*PDS*) gene in both *P. trichocarpa* × *P. deltoides* 52-225 and *P. deltoides* WV94 genomes (Figure 4A). To achieve multiplexed gene editing in poplar, we used the tRNA-based polycistronic gRNA expression system to generate two gRNAs (Figure 4B). When the construct pYL021 were tested in the hybrid poplar protoplasts, only large deletions between two targets were detected with a mutation efficiency around 31% (Figure 4B and Appendix A).

To test whether the BeYDV-derived replicons can enhance the editing efficiency, the Trex2-Cas9 expression unit and gRNAs expression unit were inserted between LIR and SIR of the BeYDV for replication (Figure 4C). In the protoplasts transformed with vector pYL023, large deletions between two target sites were also obtained (Figure 4C and Appendix A). The mutation efficiency was around 57%, which is approximately 1.8 folds of the mutation efficiency induced by the construct without the replicon (Figure 4C).

Surprisingly, for both pYL021 and pYL023 constructs, no mutations were obtained from the transient assay in WV94 protoplasts, suggesting further optimization are needed for multiplexing genome editing in this genotype.

## 3. Discussion

In *Populus* functional genomics research, *P*. *tremula* × *P. alba* 717-1B4 is perhaps the most widely used transformation model due to its high susceptibility to *Agrobacterium*. To date, the transformation methods for *P*. *tremula* × *P. alba* 717-1B4 are well-developed and optimized [38,39,40]. In general, it is simple to induce transgenic shoots selected from shoot elongation medium to rooting in root medium in *P*. *tremula* × *P. alba* 717-1B4 transformation [34,35]. In contrast, we found that the rooting rate of *P. deltoides* WV94 transformation is generally less than 10% under the same condition. The low rooting rate may be partly caused by the existence of excessive false positive shoots because the rooting rate of wild-type *P. deltoides* WV94 is typical over 50%. Therefore, the identification and selection of positive transgenic shoots prior to root induction is critical in *P. deltoides* WV94 transformation. PCR genotyping and GUS staining are commonly used for the detection of transgenic events in plants [41,42]. However, both methods can only be performed by collecting the transgenic tissue and followed by multiple-step treatments, which is relatively time-consuming and labor-intensive. In addition, GFP-based screening of transformants is less time-consuming and can be performed in a non-destructive manner [43,44,45]. However, a fluorescence microscope equipped with the specific filter is indispensable in this method.

In this study, we demonstrate that the UV-visible reporter eYGFPuv can be used to quickly select positive transgenic events in a non-destructive manner without requiring expensive equipment (e.g., fluorescence microscope) and kits. The application of eYGFPuv requires only a portable UV lamp, which is at least 100 times less expensive than a fluorescence microscope. Furthermore, the utilization of of eYGFPuv is not limited by sample size and physical location. Unlike the traditional PCR genotyping that is typically used in the late stage (e.g., root induction stage), the eYGFPuv-based selection can be performed in the early stage of transformation (i.e., shoot induction stage) (Figure 1C). Moreover, the selection can be completed in the Petri dish plate without the requirement of opening the lid, thus avoiding any potential contamination. In addition, visualization and quantification of positive transgenic events are useful to rule out the non-transformed explants and determine the transformation scale in the early stage to obtain sufficient transgenic events, saving time, materials, and effort for completing the transformation tasks. Moreover, the transgenic events can be easily tracked under different conditions regardless any container (e.g., Petri dish plate, magenta box, and pot) and location (e.g., growth room, greenhouse, and field).

Previous reports showed that antibiotic-based selection in plant transformation could result in the regeneration of escapes (i.e., false positives) and chimerical shoots [46,47]. This is consistent with our results revealing three types of the poplar shoots regenerated from antibiotic-based selection: (1) true transgenic events showing green eYGFPuv signal across the whole shoot (Figure 1C), (2) chimerical shoots showing green eYGFPuv signal in part of the shoot (Figure 2A), and (3) escapes that did not show any green eYGFPuv signal at all (Figure 1C). Therefore, the eYGFPuv reporter enables an easy detection of chimerical shoots or escapes at the early stage of plant transformation and consequently reduces the cost and time for molecular characterization (e.g., PCR-based genotyping, RT-qPCR analysis of gene expression) of a large number of regenerated shoots to identify true transgenic events with expected expression pattern of target genes. However, we should not use eYGFPuv-based early selection to fully replace PCR-based genotyping and RT-qPCR analysis of gene expression. We recommend using PCR-based genotyping and RT-qPCR analysis to further characterize the transgenic events showing expected spatiotemporal patterns of eYGFPuv signal.

The stability and reliability of eYGFPuv have been comprehensively verified in *Arabidopsis*, tobacco, *P*. *tremula* × *P*. *alba* 717-1B4, and citrus [34]. In the present study, the transgenic events selected by green fluorescence were confirmed by both PCR genotyping and GUS-staining consistently (Figure 2), indicating that eYGFPuv is highly accurate in plant transformation. Additionally, green fluorescence was visible in the very small, regenerated tissue in the shoot induction stage (Figure 1C). eYGFPuv is practically sensitive in detecting gene expression as a result. Intriguingly, events with relatively low levels of gene expression were eliminated using the fluorescence intensity seen in transgenic shoots (Figure 1C and Figure 3D).

We demonstrate that the transformation process in *P. deltoides* WV94 could be completed within 3–4 months (Figure 1), much faster than the traditional poplar procedures (6–12 months) [48]. Transgenic shoots with green fluorescence were observed in approximately 8.7% of explants (Table 1) using *Agrobacterium* strain EHA105. It has been reported that different *A*. *tumefaciens* strains resulted in different transformation efficiency in plants [49,50]. Therefore, to achieve higher transformation efficiency in the future, alternate *A*. *tumefaciens* strains (e.g., AGL1, GV3101) should be tested. Notably, the explant type also affects transformation efficiency dramatically in *P. deltoides* WV94. Unlike tobacco and *P*. *tremula* × *P*. *alba* 717-1B4 that can be easily transformed using leaf disks [34,51], we observed that the transformation of *P. deltoides* WV94 using leaf disks often results in a very low regeneration rate due to insufficient callus induction. In contrast, shoots are induced much easier from explants including stems, petioles, and the base of midrib directly [52,53]. Thus, stems, petioles, and the base of midrib are recommended as the explants for transformation of *P. deltoides* WV94.

It has been shown that incorporating the three prime repair exonuclease 2 (TREX2) exonuclease with Cas9 can enhance the efficiency of targeted mutagenesis [32,33]. Here, we demonstrated that the Trex2-Cas9 systems together with the tRNA-based polycistronic gRNA expression system could induce mutations at two target sites simultaneously. These results provide options for multiplexed gene editing in one of the two tested poplar clones, i.e., 52-225. Interestingly, we obtained large deletions between two target sites, indicating both gRNAs are effective. The results also suggest that Trex2-Cas9 can be a useful tool for generating large deletions and deleting gene clusters in poplar 52-225. Furthermore, we found that the deletion size ranges from 6 to 34 bp in the target region, which is consistent with reports in other plant species [32]. To further elucidate the characteristics of the mutation profile induced by Trex2-Cas9 in poplar in general, more target sites need to be investigated.

We correspondingly demonstrated that the BeYDV-derived replicons enhanced the multiplex gene editing efficiency, indicating BeYDV-derived replicons can also be used for optimizing the gene editing tools in 52-225. BeYDV-derived replicons have been applied for increasing precision gene targeting efficiency by providing more donor template [33]. Our results lay the foundation for further application of the BeYDV-derived replicons in precision gene editing in poplar in general.

Unexpectantly, we obtained measurable mutation efficiency from the transient assay of the hybrid poplar but did not observe any mutations from transient assays in poplar WV94. Previous studies showed that the gRNA sequence determines the mutagenesis efficiency and mutation profile [54,55]. Because the designed gRNAs can generate mutations in the hybrid poplar genotype but not in the WV94, we believe that the undetectable mutation efficiency in WV94 might be due to the chromatin structure. The efficiency of editing for different gRNAs varies greatly due to the chromatin structure, both in animal and plant cells [56,57,58]. Low mutagenesis efficiencies were mostly associated with low chromatin accessibility [59,60,61]. In the future, to apply the CRISPR/Cas9-based gene editing system to *P. deltoides* WV94, we suggest that more target sites need to be investigated and further optimization of the multiplexed gene editing system is needed.

In conclusion, the eYGFPuv system can be used to improve the established transformation protocols in *Populus* and merit testing in other model and non-model plant species [34,62]. Multiplexing genome editing using BeYDV replicons and Trex2-Cas9 system is reasonably effective in protoplasts, laying the foundation for genome editing in poplar through stable transformation. The combination of the eYGFPuv, BeYDV replicons, and Trex2-Cas9 system offers a promising approach to optimize and accelerate multiplexing genome editing in poplar.

## 4. Material and Methods

Utilizing *Agrobacterium*-mediated transformation, we transformed a vector containing eYGFPuv and GUS dual reporters into the wild type of *P. deltoides* clone WV94. Positive transgenic events were visualized based on the green fluorescence of explants. PCR-based genotyping and GUS staining were used to verify the transgenic events selected by green fluorescence. The positive rate of transgenic events in different growth stages were calculated by counting the events with green fluorescence. Meanwhile, we also examined the efficacy of CRISRP-Cas9 system in poplar through protoplast transformation. The poplar genomic DNA was extracted from transformed poplar protoplasts for PCR genotyping. The PCR products were sent for next-generation sequencing for the identification of genome-edited events.

### 4.1. Plant Materials

The wild type of *P. deltoides* clone WV94 was received form ArborGen, Summerville, SC, USA and maintained in tissue culture room. The *P. trichocarpa* × *P. deltoides* clone 52-225 was maintained in in vitro growth conditions with 16 h light/8 h dark period at 25 °C.

### 4.2. Vector Construction

The eYGFPuv/GUS dual reporter vector pAXY0003 was created by replacing the 1 × 35S promoter of 1 × *eYGFPuv* expression vector [34] with a PCR-amplified 2 × 35S promoter using the NEBuilder HiFi DNA Assembly Cloning Kit (New England BioLabs, Catalog #E5520S, Ipswich, MA, USA). The genome editing vectors were assembled using the genome engineering toolkit developed by the Voytas Lab (Saint Paul, MN, USA) [33]. One published gRNA [31] and one gRNA designed from the CRISPOR [63] were chosen for targeting the *PDS* gene in poplar. *CmYLCV* promoter, gRNAs and scaffold sequences, and terminator sequences were chemically synthesized by Integrated DNA Technology (Coralville, IA, USA) and inserted into pMOD_B0000 https://www.addgene.org/91058/ (accessed on 9 December 2022) using Gibson assembly with NEBuilder^®^ HiFi DNA Assembly master mix (NEB, Cat. No. E2621). Synthesized 35S promoter, *eYGFPuv* coding sequence, and HSP terminator sequence were inserted into pMOD_C0000 https://www.addgene.org/91081/ (accessed on 9 December 2022) using Gibson assembly method. The Trex2-Cas9 expression module, pMOD_A0902 https://www.addgene.org/91026/ (accessed on 9 December 2022) together with the gRNA expression module, pYL015 and the *eYGFPuv* expression module, pYL009 were cloned into T-DNA backbone by Golden Gate cloning strategies [33]. Plasmids sequences were confirmed by Sanger sequencing. Plasmids are listed in Appendix A, and their corresponding DNA sequences are available at Addgene https://www.addgene.org/ (accessed on 9 December 2022).

### 4.3. Genotyping of Transgenic Plants

The leaves from transgenic plants were collected and used for genomic DNA extraction following a protocol described previously [64]. Transgenic plants were PCR genotyped using GoTaq^®^ Master Mixes (Promega, Madison, WI, USA, Cat. No. M7122). We used 1 μL of genomic DNA as a template with the following PCR cycling conditions: 95 °C for 2 min, 35 cycles of 95 °C for 30 s, 55 °C for 30 s, and 72 °C for 1 min, with the final elongation step at 72 °C for 5 min. Primers for genotyping (Genotype_F and Genotype_R) have been listed in Appendix A.

### 4.4. GUS Staining

The leaves from transgenic plants were collected and used for GUS staining, with a wild type as the control. GUS staining was performed using the β-Glucuronidase Reporter Gene Staining Kit (Sigma-Aldrich, Darmstadt, Germany) following the instructions. Transfer plant tissue to a 1.5 mL or 50 mL tube. Add staining solution to the tube and make sure that the tissue is covered with the solution. Close the lid tightly and incubate at 37 °C for up to 24 h. A blue stain develops with time. When expression is high, the solution becomes blue due to leakage of the blue reaction product from the tissue.

### 4.5. Protoplast Transformation

The protoplasts were isolated from poplar clones WV94 or 52-225 and transformed with PEG/Ca^2+^ methods as previously reported [65]. In brief, leaves from one-month-old *P*. *deltoides* WV94 and *P*. *trichocarpa* × *P. deltoides* clone 52-225 plants were sliced into strips and digested with enzyme solution (0.4 M mannitol, 20 mM KCl, 20 mM MES, 10 mM CaCl_2_, 5 mM β-mercaptoethanol, 0.1% BSA, 0.8% macerozyme R10, and 3% cellulase R10). After 3–5 h, the protoplasts were washed with W5 solution (154 mM NaCl, 125 mM CaCl_2_, 5 mM KCl, and 2 mM MES). We transformed 20 μg plasmids of pYL021, pYL023, and pGFPGUSPlus (negative control) [66] into 40,000 cells using PEG/Ca^2+^ solution (100 mM CaCl_2_, 0.2 M mannitol, 40% PEG4000) and cultured for 48 h in dark conditions at room temperature. After washing with W5 solution, the protoplasts were collected for DNA extraction.

### 4.6. Mutation Profiling

DNA was extracted from protoplasts using modified SDS method [67]. Primers (oYL_063_Deep_seq_F and oYL_062_Deep_seq_R) for amplifying PDS targets and next-generation sequencing are listed in Appendix A. Q5 high-fidelity polymerase (New England Biolabs) was used for amplifying the target DNA region with the following PCR cycling conditions: 98 °C for 30 s, 35 cycles of 98 °C for 10 s, 65 °C for 30 s, and 72 °C for 30 s, with the final elongation step at 72 °C for 2 min. Next Generation Sequencing (NGS) was used to sequence amplicons via GENEWIZ Amplicon-EZ services. Mutations were assessed for each sample using Cas-Analyzer [68]. Minority read sequences represented less than 10 times were considered background.

### 4.7. Visualization of Transgenic Events

The transgenic plants were visualized using an Ultraviolet A (UVA) flashlight LIGHTFE UV302D (365 nm) (TURBO LIGHTFE, Zhongshan, China) and the method for visualization has been fully described previously [34]. The images were taken using iPhone 11 (Apple, Cupertino, CA, USA).

### 4.8. Poplar Transformation

The eYGFPuv/GUS plasmid pAXY0003 was transformed into *A*. *tumefaciens* strain ‘EHA105’ using electroporation and then transformed into the *P*. *deltoides* WV94 following the steps described in 2.1. In step 2, the OD of *A*. *tumefaciens* was adjusted to 0.8~1.0 and acetosyringone (20 µM) was added to promote transformation. In step 4, sterile water containing Timentin (200 mg/L) and cefotaxime (300 mg/L) was used as washing solution. In the selection medium including SIM94, SEM94, and RM94 [69], kanamycin (100 mg/L) was used to select transgenic events and Timentin (200 mg/L) and cefotaxime (300 mg/L) were used to inhibit the growth of *A*. *tumefaciens*. eYGFPuv fluorescence was checked regularly using a 365 nm UV flashlight.

## Figures and Tables

**Figure 1 plants-12-01657-f001:**
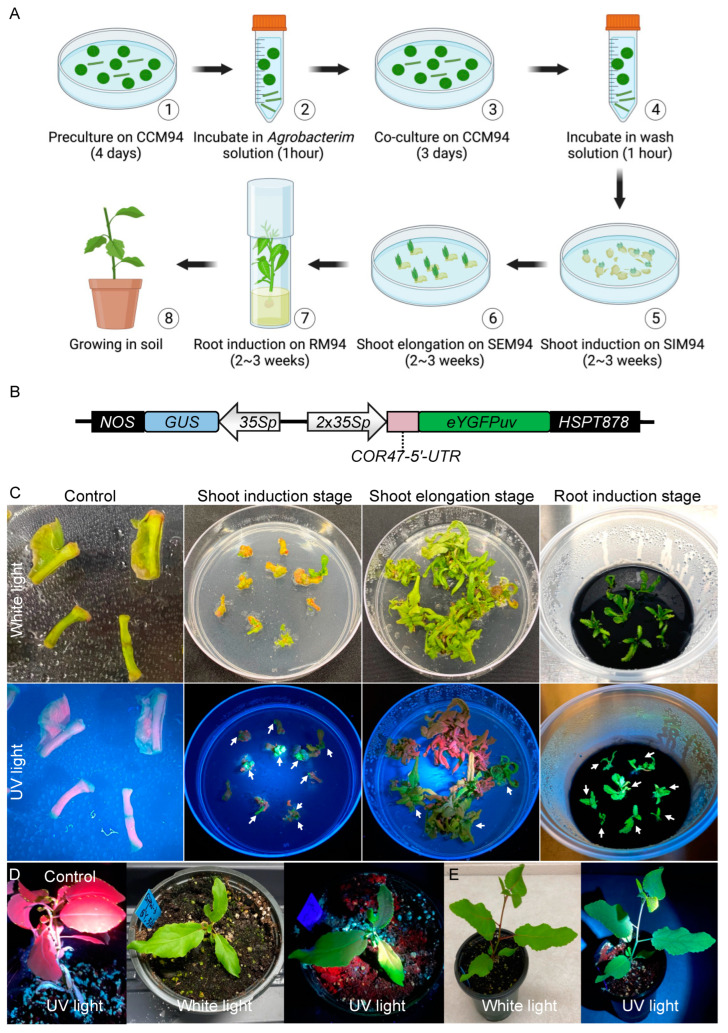
eYGFPuv-assisted plant transformation in *Populus deltoides*. (**A**) The procedures of tissue-culture-based and *Agrobacterium*-mediated plant transformation in *Populus deltoides*. (**B**) Illustration of eYGFPuv/GUS dual reporter vector (pAXY0003). 35S; *Cauliflower mosaic virus* (*CaMV*) 35S promoter; *NOS*, the nopaline synthase terminator; *HSPT878*, *HSPT878* terminator. (**C**) Visualization of eYGFPuv in different stages of plant transformation under UV light. White arrows indicate transgenic shoots with green fluorescence. Control indicates the transformation using a vector without the *eYGFPuv* gene. (**D**) Visualization of eYGFPuv in poplar plants, six weeks in soil. Control indicates the wild-type plant under UV light. (**E**) Visualization of eYGFPuv in poplar plants, nine weeks in soil.

**Figure 2 plants-12-01657-f002:**
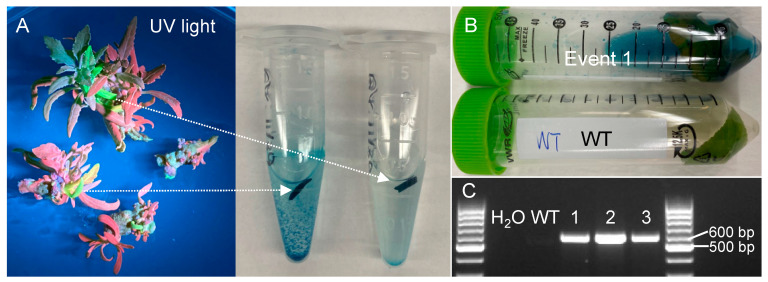
Verification of transgenic events selected based on *eYGFPuv* expression. (**A**) GUS staining of leaf samples collected from shoot elongation stage. (**B**) GUS staining of leaf samples collected from nine-week-old plant in soil. (**C**) PCR genotyping of transgenic events with green fluorescence.

**Figure 3 plants-12-01657-f003:**
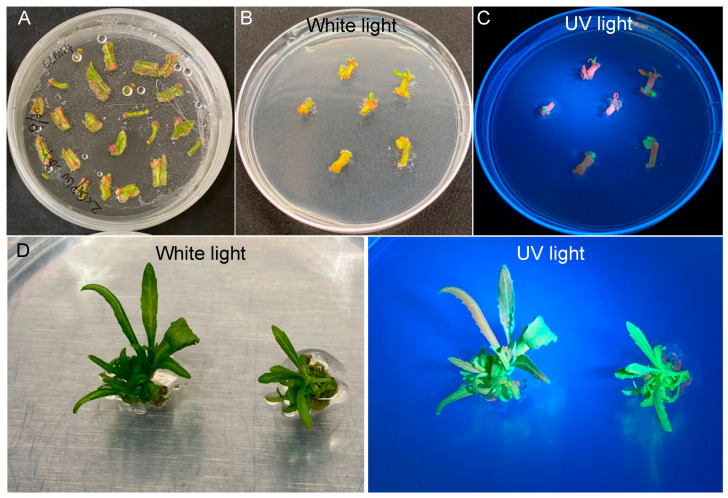
Quantification of transformation efficiency in different stages of plant transformation. (**A**) The explants survived on selection medium. (**B**) The explants with shoot induction. (**C**) The explants with at least one GFP callus. (**D**) The explants with at least one GFP shoot.

**Figure 4 plants-12-01657-f004:**
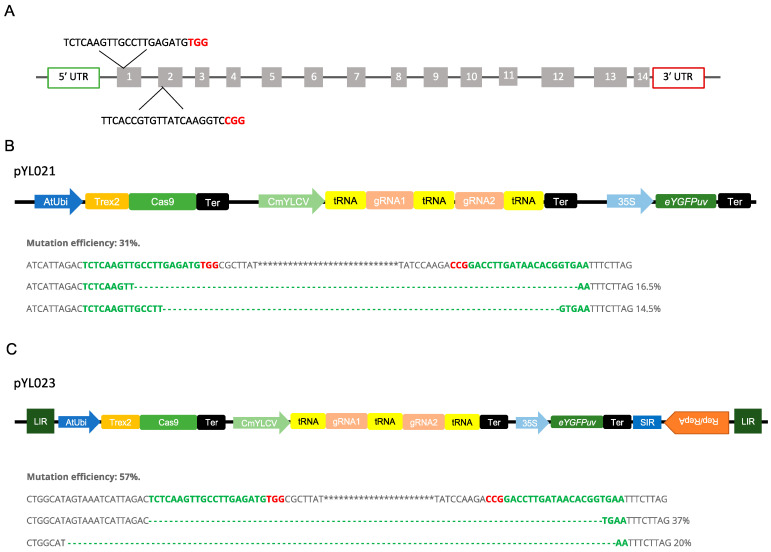
Vectors design and mutation efficiency for multiplexing genome editing in *Populus*. (**A**) Target regions of the *Phytoene desaturase* (PDS) gene and sequences of two gRNAs. Bold red characters represent PAM sequences. (**B**) Vector design for pYL021 and mutation efficiency determined by NGS in *P. trichocarpa × P. deltoides* (clone 52-225) protoplast. Red characters represent PAM sequences and dash lines represent deletions. (**C**) Vector design for pYL023 and mutation efficiency determined by NGS in *P. trichocarpa × P. deltoides* (52-225) protoplast. Red characters represent PAM sequences and dash lines represent deletions. LIR, large intergenic region; SIR, short intergenic region; Rep, replication-initiation protein; *tRNA*, 77-bp pre-*tRNA^Gly^* gene; *CmYLCV*, *Cestrum yellow leaf curling virus* promoter; *AtUbi*, *Arabidopsis* ubiquitin promoter; *Trex2*; the three prime repair exonuclease 2 exonuclease; *Ter*; terminator; 35S; *Cauliflower mosaic virus* (CaMV) 35S promoter.

**Table 1 plants-12-01657-t001:** Quantification of transformation efficiency during plant regeneration.

Name	Quantity of Explants	Efficiency
Total explants	150	N/A
Survived explants	143	95.3%
Explants with at least one shoot	80	53.3%
Explants with at least one GFP callus	23	15.3%
Explants with at least one GFP shoot	13	8.7%

## Data Availability

The plasmids are available at Addgene. Disclosure: This manuscript has been authored by UT-Battelle, LLC under Contract No. DE-AC05-00OR22725 with the U.S. Department of Energy. The United States Government retains and the publisher, by accepting the article for publication, acknowledges that the United States Government retains a non-exclusive, paid-up, irrevocable, worldwide license to publish or reproduce the published form of this manuscript, or allow others to do so, for United States Government purposes. The Department of Energy will provide public access to these results of federally sponsored research in accordance with the DOE Public Access Plan (http://energy.gov/downloads/doe-public-access-plan).

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
