# Peer review of "eYGFPuv-Assisted Transgenic Selection in *Populus deltoides* WV94 and Multiplex Genome Editing in Protoplasts of *P. trichocarpa* × *P. deltoides* Clone ‘52-225’"

_plants, 2023, doi:10.3390/plants12081657_

Round 1
Reviewer 1 Report
The manuscript entitled “eYGFPuv-assisted plant transformation and multiplex genome editing in Populus'', studies the effects a simple and reliable method for early screening of transgenic events in Agrobacterium-mediated transformation of WV94 through the application of the eYGFPuv reporter gene; as well as we the genome editing using BeYDV replicons and Trex2-Cas9 system in protoplasts. The authors presented that the BeYDV-derived replicons can enhance the multiplex gene editing efficiency, indicating BeYDV-derived replicons can be used for optimizing the gene editing tools in poplar.
Major suggestions:
The authors need to revise the title of the paper in a more meaningful way. the title is long and has some unnecessary information, suggestion: “Transformation and multiplex genome editing in Populus”;
Please improve the abstract to cover the important topics reviewed and discussed in this article. The abstract is written in a way lacks logic. It should highlight the salient findings more critically;
Keywords are present in the title (plant transformation; eYGFPuv; genome editing), choose other indexing terms for the article;
In the text, reference numbers should be placed in square brackets [ ], and placed before the punctuation; for example [1], [1–3] or [1,3];
The results of this study are not fully explained therefore the interpretation of the results is very difficult. The author needs to provide the % increase or decrease rather than just writing ''significantly increased….''. Authors should discuss the results integrally. The discussion is based on individual results. I suggest that integrating the results will give more value to the work. I suggest that you discuss by integrating all your results.
The discussion is poorly written hence, needs rewriting. The discussion is it's short and poor. The discussion should be further strengthened by adding some more relevant papers. The literature search is insufficient, only few related research papers in the past five years are cited (44.1%, approximately), add the latest research results appropriately.
The report on M&M is very succinct! I suggest a better description of the experiments carried out, such as: Genotyping of transgenic plants; GUS staining; Protoplast transformation; Mutation profiling; and Visualization of transgenic events;
Provide experimental work plan at the start of M&M. No detail description is available about the experimental design. What statistical method is used?
Minor suggestions:
Lines 35-36: “More recently, the use of Populus spp. as a source of renewable bioenergy has also been widely highlighted.” In what way specifically?
Lines 57 to 59: “In comparison with the transformation in P. tremula x P. alba clone 717-1B4, WV94 usually displays a shorter transformation cycle but at lower transformation efficiency (data not shown).” If it does not show results or any references, I suggest removing this sentence.
Lines 121 to 124: I suggest redoing the figure 1B, it was not clear in the figure.
Lines 128 to 129: Green fluorescence was not very evident in shoot elongation stage to root induction stage. If there is a better figure, I suggest the exchange.
Line 181: Exchange 1.8 times for 1.8 folds.
Reviewer 2 Report
thanks to the authors for the work done on this manuscript
General note, genes name must be written in italic
Line 80:
The first appearance in the introduction of the abbreviation of eYGFPuv gene is in line 80. Please add the full wording of the gene when it appears the first time in the introduction.
In line 93
The Author described the incubation of explants with Agro solution including antibiotics. Please specify which antibiotic type. Logically it is an antibiotic against falls agro cells such as spectinomycin or streptomycin extra. But the reader might be confused and think it is a plant selection antibiotic because the author discussed in former lines the selection problems resulting in false positive plants.
Generally, the transformation protocol is poorly described. A lot of information is missing which is essential to repeat this work by other groups interested in Poplar trafo. Please specify these points:
Which Agrobacterium strain was used and its reference if possible?
What is the OD of bacterial cells in the transformation step and the time of incubation?
Which antibiotic is used to screen the agrobacterium and what is the concentration?
What is the washing solution?
Include the composition of SIM94, SEM94, and RM94, or add a reference for them.
In line 110:
The author commented on the transformation efficiency of different explants in a very plausible way without showing any experimental setup such as explants number, regeneration number, or how they calculated for transformation efficiency.
Line 112:
What is the concentration of acetosyringone?
Line 114:
The author mentioned that high selection increased transformation efficiency. What does mean high efficiency, what is the percent of the transformation rate? What are the different concentrations of kanamycin that have been used? And why they selected 100mg/l? please include these results. With pictures of kanamycin's effect on the vigor and viability of regenerated shoots.
Line 127:
In figure 1C, the given images did not show any green fluoresce as the author mentioned, I did not see it. Can the author indicate it by arrows where the GFP signal is visible? The author should explain how the transgenic plantlets were exposed to UV light and how they were imaged. What are the UV light type and wavelengths? Which camera was used to record these mages?
This experiment is not informative at all, it lacks a control where non-transgenic explants were UV exposed and imaged. This is important to compare with what is claimed to be transgenic. The author must include control images.
The author should discuss why UV method is better than using a fluorescent Binuclear-microscope. Where GFP-expressing plants can be non-invasively tested for GFP expression. The presented UV method seems to lack accuracy because a weak GFP signal will be undetectable. Can the author comment on how accurate and sensitive is the UV method?
The author should validate using UV method before confirming a transgenicity. To do so, the author should compare the results of UV-positive plants with confocal laser scanning microscopy and GFP-PCR test. This comparison will show how reliable is the UV test.
To further confirm transformation, PCR test is not adequate and a southern blot should be performed.
The PCR results in figure 2C lack the presence of positive control which is the pDNA fragment from the construct used for transformation. Would the author include pDNA in the PCR?
In the protoplast assay: the author found mutations in the hybrid poplar protoplasts while they could not detect any in the WV94 genotype. This might indicate sequence differences between selected target motifs of both genotypes (the hybrid poplar and WV94). The author might amplify the target region including target motif1 and target motif2 of both genotypes and align the sequencing together. It might be the case that the target motif including PAM has snips
Discussion: the author discussed the importance of the non-invasive, however, did not discuss that this method is only useful when establishing transformation protocol in a new species. And in cases where transformation does not include a fluorescence marker, the method will not be useful.
Reviewer 3 Report
The present manuscript “eYGFPuv-assisted plant transformation and multiplex genome editing in Populus” presented a work-related Populus species which is recalcitrant to traditional transformation. The authors made transgenics and tried to perform genome editing. The research work presented in the manuscript is fair enough to be published in the journal plants. However, the authors should address some of the comments given below.
Comments:
Make the abstract short and remove the lengthy introduction part from the abstract.
Change the Fig. 3 table screenshot into a proper expression graph
Discuss why authors could not detect the mutations in genome-editing plants.
Round 2
Reviewer 2 Report
Thanks to the authors for considering and answering my former question and concerns. Transformation of plant trees is indeed a challenging approach however it is of very high importance. Providing a successful and reproducible protocol is in high demand by the Poplar tree research and business community. I highly appreciate the efforts given by the author however, I see the efforts made to conclude this manuscript were not enough to make it published for the following reasons.
1- The author claimed on several occasions throughout the manuscript, that using UV-visible eYGFP is an alternative way to PCR, southern blot, or RNA methods, which is not true because transgene integration must be confirmed by T-DNA integration in the target plant genome. This can be done only through PCR, Southern blot, or RT-PCR. The proposed fluorescent method can only confirm the gene expression which can be either transient expression or clonal expression (chimeric) rather than stable transformation. Stable transformation means the ubiquitous integration of T-DNA as well as the inheritance of the T-DNA from one generation to the next.
To conclude, the main message of the manuscript is wrong, the UV method is useful to detect transient gene expression and early steps of stable transformation but will not confirm T-DNA integration and inheritance of the transferred T-DNA. The manuscript would be useful for the plant transformation community if it is proposed as an early and easy way to screen out the false positive plants but not to confirm the stable transformation.
2- The author was not careful enough in writing this manuscript. Simple mistakes like not writing scientific names of species and plant genes in italic.
3- English needs improvement
4- The author claimed on several occasions in the manuscript that a big advantage of the proposed UV method is that, it is a non-invasive GFP detection method. However, fluorescence detection under a simple UV binuclear microscope is also non-invasive detection that is being used over many decades in detecting reporter genes in plants and other organisms.
5- The author did not discuss all the chimeric structures (except one time in line 225) which are visible in all provided examples in this manuscript. it is important to discuss how it is possible in this protocol to regenerate transgenic plant chimerism free.
Comments on the manuscript:
1. In the title, please indicate that the genome editing is done in protoplasts only. The current title is misleading and overselling the topic.
2. In the abstract, the author wrote “ Here, we first optimized the protocol for Agrobacterium-mediated stable transformation in P. deltoides WV94 using a UV-visible reporter called eYGFPuv.” The proposed UV method does not affect Agrobacterium-based transformation. The author probably means to improve the detection of GFP transgenic plants but not improve Agrobacterium efficiency. This is misleading and the reader might think the UV method enhances agrobacterium transfection.
3. in results, lines 90 to 99, this is all materials and methods and should be transferred to the materials and methods part.
4. In line 103, the author described the transformation efficiency of WV94 is relatively low. Can the author be specific and show accurate data in numbers or percentages?
5. Between lines 121 and 131, the author discussed the selection based on the green signal and indicated that positive explants showed a green signal and negative explants showed a red signal. However, the author did not comment on the chimeric situations in Figure 1C second, third, and fourth photos where both red and green signals were visible. In addition, at the borders of the control explants there is a green signal, can the author explain the presence of this signal in wild-type explants?
6. In general, the images in Figure 1 are of very poor quality to confirm the green signal of positive plants particularly in 1D and 1E. the green signal is not visible in 1D and 1E.
7. Did the author use intronized eYGFP and GUS genes? The vector map in Figure 1 B did not show whether GUS and GFP gene includes introns. if the genes do not include an exon then the bacteria can express them and it will be not possible to distinguish between the signal that comes from the bacteria or from the transgenic plant. Agrobacterium cells that were not removed by washing or killed by antibiotic still can grow on the explants and can express non- intronized genes and the green signal or the GUS signal can also come from bacterial expression, not from the explant expression.
8. The name of genes on the vector map in Figure 1 B must be written in italic.
9. Figure 1A and 1B should be moved to materials and methods section they are not part of the results.
10. In figure 2A, the green signal is distinguishable from the red which indicates a high chimeric situation on the same shoot, meaning in the same shoot it is found green leaves and red leaves. How the author can confirm stable transformation with such very high chimeric situation? It looks like a clonal transformation in a few leaves rather than a complete transgenic shoot.
11. In figure 2C the PCR still did not include a positive control.
12. In line 157 the author mentioned the kanamycin concentration is 100µg/l which thousand times less than the normal concentrations used for selection. is it a typo mistake or why the author used very low concentration?
Discussion:
13. Line 202, the author discussed low rooting capacity in P.deltoides WV94 might be due to high rate of false positive. This can be tested easily by showing regeneration data without transformation. If the rooting is poor in regeneration experiments without transformation, this means that this is the capacity of the genotype and transformation is not affecting rooting capacity.
14. in Materials and Methods,
15. the author did not mention any information about how gRNAs were designed or how to target motifs were selected. Which platforms were used to perform this part? This should be explained in materials and methods.
16. Protoplast assay was explained briefly, it is recommended to cite a paper where the protocol was adopted from it.
17. The UV method can only show the expressed protein but cannot tell about gene integration in plant genome since the GFP can be also expressed by agrobacterium.
18. Lines 95 to 95: in response to my former interrogation the author kindly added the antibiotics used during the washing. The author claimed using Timentin and Cefotaxime kill the Agrobacterium. The Author wrote, in the comments, washing for 1h while did not add it in the manuscript, please add it.
19. Is 1 hour is sufficient to kill all Agrobacterium cells? Did the author performed any test to confirm the death of agrobacterium?
20. One hour is known to be insufficient to kill all bacterial cells. The survived bacteria will later recover and propagate during the culture. Having the Agrobacterial cells on the callus or shoots will mislead the conclusion, because the bacterial cells can express the eYGFPuv gene and its DNA can be detected by PCR as well as the GFP protein in fluorescence microscope. In this case, it is impossible to distinguish whether GFP expression come from the bacterial cells or transformed plant cells.
21. in line 360, please write the A. tumefaciens in italic
22. In response to my former interrogation in point 6: the author claimed adding the concentration of acetosyringtone but I did not find it. Can the author recheck?
23. In response to my former interrogation in point 9: the author added a negative control explants. Under the UV light, a greenish color is visible at the border of the explants. Can the author explain this greenish signal?
24. In response to my former interrogation point 13, the author claimed the stability of the method and did not show the positive control in the figure 2c. the author answer is insufficient and a positive control has to be added to the manuscript to have proper comparison. However, the author sent in his comments a gel picture that does not have any bands, what is that?
Round 3
Reviewer 2 Report
Point 1: Thanks to the author for answering the questioned points. As mentioned in my former review, the transformation of plant trees is very challenging and any progress would be of great help to the research and business community of poplar trees and probably other species as well. My main question to the author is not yet being answered, what is new in this manuscript to improve transformation? Of course, providing a successful selection method is very useful, however, the mentioned method alone is not sufficient and the author should emphasize performing a PCR to confirm T-DNA integration in parallel to the fluoresce method. In contrast to my former comment, the author mentioned in lines 21 – 23 in the abstract, that the fluorescence method is enough without the requirement of the characterization of the target gene at the DNA or mRNA level. Confirming transformation must include T-DNA integration and gene expression. This statement in lines 21-23 misleads the readers in particular young scientists.
Point 2: In addition: the author mentioned in the transformation protocol some changes such as adding the acetosyringon, washing steps, and extra without showing any comparison. For example, if applying different concentrations of acetosyringon the comparison showed be shown in the results. What the author showed is a qualitative measure, which is not enough for publishing and it is rather a lab protocol. The author should quantitatively present his data with statistically relevant replications.
Point 3: The title of the manuscript should include and highlight the main achievement of the manuscript which is in my opinion, improving of transgene selection (not plant transformation) and confirming genome editing in a protoplast essay. Thanks for adding the word transient in the title but it is not used correctly, because genome editing itself is not transient but the Cas9 gene is transiently expressed. This transient expression is enough to generate genome editing even though it is clear that there are no plants that will be regenerated from the modified protoplast.
Still open questions from my former review:
Point 4: related to point 1 in my former review: The author replied, that they did not mention that the eYGFPuv is an alternative way to PCR. Please revise your statement in the abstract between lines 21-23. When I mentioned in point 1 that PCR, southern blot, and extra are required, the author misunderstood the meaning. I mentioned them to give an example of what can be done, and again if the author revised his statement in lines 21-23 he will find that they meant clearly that the fluorescence method is enough and no need for other methods (line 22). In the same line, the author stated that the characterization of the target gene at the DNA level is time-consuming which is not true at all and misleading. Nowadays, genomic DNA isolation kits are available, one can in one hour isolate up to 96 samples in a 96-well DNA purification plate, and there are several commercial products available in the market. Then the PCR and gel running and imaging require another couple of hours. The authors should confirm every transgene event by PCR as they did in lines 239-240. The advantage of using the fluorescence method is to reduce the number of plants to be PCR screened not to stand-alone to confirm T-DNA integration.
Point5: related to point 2 in my former review: the author indeed changed several gene names to italics but not all. As an example of yet not in italics, in Figure 1B, terminator names should be in italics, in line 139 eYGFP/GUS, 35S, CaMV and NOS, line 162 eYGFPuv, and this is not all, please revise carefully.
Point 6: related to point 4 in my former review: Thanks to the author for listing the advantages of using the UV lamp method. The author should refine the advantages and add them to the discussion. I just disagree that using a simple fluorescent Bi-nuclear microscope requires a time of training. This is not true, in a few minutes, it is possible to learn and use.
Point 7: related to point 5 in my former review, the author in his response stated that regeneration from a single cell cannot produce chimerism, which fully true. However, the author does not have any proof that the elaborated plants regenerants or shoots originated from a single cell. When starting a culture from a tissue-explant and producing callus to my knowledge there is no method to track the origin of a shoot. It is only possible to speak about the single-cell origin in a few cases such as pollen culture, ovule culture, and cell suspension. The comment of the author is again misleading.
Point 8: related to point 8 in my former review: Materials and methods must be given in their proper section of the manuscript and cannot be placed in the results section, this is wrong.
Point 9: related to point 10 in my former review, I made a screenshot of figure 1 and indicated the green part in WT (control) with green arrows while indicating the non-transgenic parts of chimeric structures with red arrows. Based on these pictures, can the author comment on how specific and reliable is the UV-lamp method? These pictures show how UV-lamp is not precise enough and performing a PCR, in addition, is a must.
Point 10: The author kindly added control in Figure 1D, it would be more systematic to move the control picture to the left side of the figure then all controls to are on the left side under each other.

Point 11: related to point 13 in my former review; the author indeed changed several gene names to italics but not all. As an example, in figure 1B, terminator names should be in italics, in line 139 eYGFP/GUS, 35S, CaMV and NOS, line 162 eYGFPuv. Please revise properly.
Point 12: related to point 14 in my former review; it is not professional that M&M data are shown in results. They must be moved to M&M section. Please revise the basic roles of scientific writing.
Point 13: related to point 15 in my former review; The response of the author is not proper. Figure 2A is a very clear example of what chimera means. In this particular example, it is not possible to isolate a shoot without a red signal and either GUS or PCR is selective since you just collect the green leaves as indicated by arrows. I disagree about the author's response, and still cannot understand how possible to get a complete transgenic plant from this chimeric structure.

Point 14: related to point 18 in my former review; thanks to the author for answering the question. I suggest including the answer in the manuscript accordingly.
Point 15: related to point 27 in my former review; I still see green color at the border of the explants and I indicated them by green arrows.

Point 16: related to point 28 in my former review: Below is a screenshot of the author's response last version, there are no bands. this might be a technical problem. Now with the new version of the author's comments, it is fine.
